# Dated Data: Tracing Knowledge Cutoffs in Large Language Models

**Jeffrey Cheng    Marc Marone    Orion Weller**
**Dawn Lawrie    Daniel Khashabi    Benjamin Van Durme**

Johns Hopkins University

jcheng71@jhu.edu

## Abstract

Large Language Models (LLMs) are often paired with a *reported cutoff date*, the time at which training data was gathered. Such information is crucial for applications where the LLM must provide up-to-date information. However, a reported cutoff only scratches the surface. Do all sub-resources in the training data share the same cutoff? Does the model's demonstrated knowledge for these sub-resources closely align to their cutoff? We define the notion of an *effective cutoff*, which is distinct from the LLM's reported cutoff and differs between sub-resources. We propose a simple approach to estimate effective cutoffs of an LLM on the resource-level by probing across versions of the data. Crucially, our method does not require access to a model's pre-training data. Through our analysis, we find that effective cutoffs often drastically differ from reported cutoffs. To understand the root cause of this observation, we conduct a large-scale analysis on open pre-training datasets. Our analysis reveals two reasons for these inconsistencies: (1) temporal misalignments of CommonCrawl data due to non-trivial amounts of old data in new dumps; and (2) complications in LLM deduplication schemes involving semantic duplicates and lexical near-duplicates. Overall, our results show that cutoffs are not as simple as they have seemed and that care must be taken both by LLM dataset curators as well as practitioners who seek to use these models. We release our results and the code to replicate them at https://github.com/nexync/dated_data/.

## 1  Introduction

Many Large Language Model (LLM) creators do not elect to release their training data due to competitive reasons. In place of providing the exact pre-training data, they often provide a *reported cutoff* date for their model. When faced with a description that states, e.g.,"this model has a cutoff date of March 2024," does that mean all of its included resources share the exact same cutoff date? Even if the model provides an explicit reported cutoff for a resource (e.g. the Wikipedia dump date), does that imply that the model's knowledge of that resource, or *effective cutoff*, is the same as the reported cutoff date? For LLM users, these questions can be crucial: imagine a layperson using an LLM for tax advice, without realizing that the effective cutoff of the tax code is 2022 and thus outdated – despite the fact that the reported cutoff is advertised as 2023 (Fig. 1).

As a result, there has been a push for researchers to document their data (Mitchell et al., 2018; Pushkarna et al., 2022; Gebru et al., 2021; Luccioni et al., 2022), identify what data is in these models with membership inference tests (Carlini et al., 2022; Piktus et al., 2023; Marone & Van Durme, 2023), and otherwise reproduce data from their training set (Carlini et al., 2021; Ippolito et al., 2022; Nasr et al., 2023; Weller et al., 2023). However, these tests generally only check for static inclusion of the data, rather than identifying when the resources stopped being included. Even more difficult are cases where there exist *multiple* versions of a resource, where different versions can contain information that is updated, deleted, or even conflicting with the previous versions.

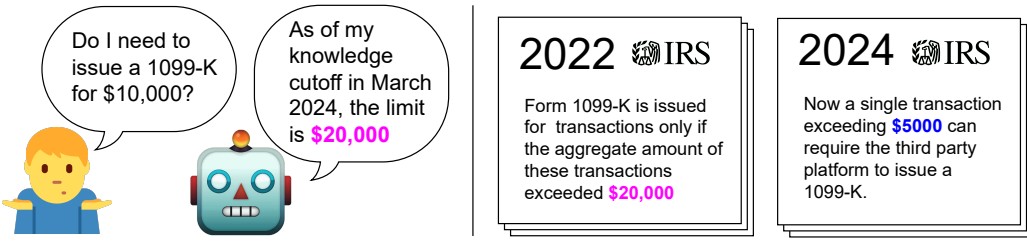

Figure 1: LLMs may contain different versions of a dataset in their training data than what is specified in a "cutoff" date, misleading users and causing potential errors.

Given the crucial importance of these knowledge cutoffs and the lack of transparency from LLM creators, we seek to automatically determine the effective cutoff of models with respect to a given resource, without needing access to the model's training data. We measure the perplexity of LLMs over varying versions of resources, identifying the effective cutoffs as the minima of the perplexity over time measurements. Our contributions are as follows: (1) we collect resource sets spanning long time frames and propose a simple method to determine effective cutoff of LLMs; (2) we show that for a variety of models (particularly newer models), the resource-level effective cutoffs differ drastically from their reported cutoff date; and (3) we provide an analysis detailing the causes of these misalignments, showing that pre-training datasets suffer from deduplication complications and that CommonCrawl dumps exhibit temporal misalignment from the dump dates.

## 2    Related Work

**Documenting and Describing LLM Training Data**    As the size of the data in LLMs increases, there have been many calls for researchers to document datasets through additional *documentation artifacts*. These include *Model Cards*, *Datasheets*, and *Data Cards* – each focusing on documentation of a specific part of a model or data source (Mitchell et al., 2018; Gebru et al., 2021; Pushkarna et al., 2022). The open-access community has adopted versions of these (e.g. Huggingface model descriptions) but they do not provide fine-grained versioning that enables precise tracking of cutoffs. For example, it may not be clear whether scraped Common Crawl data contains additional versions of scraped Wikipedia.

Other research has focused on more fine-grained analysis, such as the role of filtering in LLM-data creation (Gururangan et al., 2022; Lucy et al., 2024) or how PII, toxic data, n-grams, and provenance play a role in LLM data (Dodge et al., 2021; Elazar et al., 2023). These works have provided great insight into LLMs, but necessarily depend on the data being available to the public. In contrast, our work focuses instead on determining what temporal versions of data exist in a model, *without* access to the training data.

**Membership Inference**    Many prominent LLMs do not provide their training data or even descriptive information about them, leading people to wonder if their data is included in the model's training set. Techniques like membership inference attacks (Shokri et al., 2017) have been applied to LLMs. Many strategies have been proposed for this problem: they include using similar but synthetic data (Mattern et al., 2023), prompting calibration (Fu et al., 2023), and a variety of other techniques (Hisamoto et al., 2019; Shi et al., 2023; Faysse et al., 2024). While most strategies rely on the LLM's perplexity over potential data instances, there has also been effort in black-box attacks using cloze tasks (Chang et al., 2023). Overall, membership inference testing focuses on whether an instance was included in the LLMs training set (with the critical assumption that there was only one version). In contrast, our work focuses on *which version(s)* of the data was included in the LLMs training data.

**Continual Learning in LLMs**    New written knowledge increases every day, but language models remain static. As re-training a LLM is prohibitively expensive, it is infeasible for LLMs to keep up with living online resources.[1] Thus, there exists a large field of research on continual learning, or helping LLMs stay up to date without expensive re-training. This

---

[1]For example, Wikipedia gets edited about every 2 seconds

typically involves modeling approaches that perform limited continued training to keep model knowledge up to date (e.g. Hu et al. 2023; Kasai et al. 2023, among others). Our work also involves examining the temporal knowledge in these models, but differs by focusing on if there is temporal misalignment in the original static models and what knowledge they contain rather than techniques to align them.

## 3 Methodology

We seek to probe LLMs to determine their resource-level effective cutoffs, defining the effective cutoff date of a model with respect to a resource as the date of the version of the resource that is most closely aligned with the model. This effective cutoff date can differ from the inclusion date of a model's sub-resources that LLM creators sometime provide, which is typically the *last* timestamp of that resource but does not address additional sources of earlier data.[2] This is relevant because given that a certain Wikipedia dump is included in training data, it is reasonable to assume that the effective cutoff for those articles is the corresponding dump date. However, older versions of similar text may be present in web scrapes like CommonCrawl.

To perform our analysis, we construct long spanning (2016-2023) datasets and measure perplexity on the data across a variety of models. We then analyze the implications of these *perplexity-at-time* curves and verify our results with the ground truth pre-training data.

### 3.1 Time-Spanning Datasets

Broadly, online resources included in LLM training data can be divided into three subsets: resources that involve frequent updates (e.g. legal texts or Wikipedia), resources that are static but build over time (e.g. blogs or news articles), and purely static resources (e.g. books). We construct representative datasets that allow us to probe the first two types, choosing Wikipedia for the updating resource and New York Times for the building resource.

**WIKISPAN** Wikipedia is commonly used in pre-training (Gao et al., 2020), provides broad topic coverage, and changes frequently. To create the dataset, we collect the 5000[3] most edited documents by number of edits.[4] For each of these, we use the Wiki API to collect a version of the document at monthly intervals, from April 2016 to April 2023. Our final dataset therefore consists of the same 5000 documents changing monthly over this seven year period. We call this dataset WIKISPAN, and individual documents are the version of the Wikipedia document on topic $T$ at month $M$.

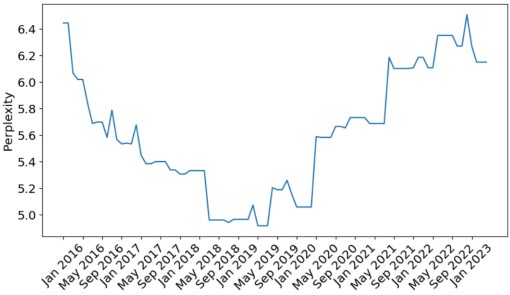

Figure 2: Perplexity of the Wiki document "Liverpool" under Pythia. Each point is the perplexity of the document at that time.

**NEWSSPAN** We use New York Times (NYT) articles to represent our building-over-time resource, as the documents contain long-form high quality text, are frequently included in pre-training data and CommonCrawl dumps, and provide a long and steady stream of new documents. For our probing dataset, called NEWSSPAN, we collect all the articles with top level domain "nytimes.com" from a curated collection of the 20 most recent CommonCrawl dumps (Soldaini et al., 2024). We bucket the articles by month according to their publication date, and collect 500 articles from each bucket from January 2016 until July 2020. Note that due to copyright concerns, CommonCrawl removed NYT articles from recent dumps

---

[2]https://huggingface.co/allenai/OLMo-7B reports using the January 2023 dump of Wikipedia
[3]We filter out 100 topics that did not exist all the way back to 2016.
[4]https://en.wikipedia.org/wiki/Wikipedia:Most_frequently_edited_pages, from May 2023

| Model Name | Pile | C4 | RW | CC Dumps | Wiki Dump | CC Cutoff |
|---|---|---|---|---|---|---|
| Pythia (Biderman et al., 2023) | ✓ | | | | | ? '20 |
| GPT-Neo (Black et al., 2022) | ✓ | | | | | ? '20 |
| GPT-J (Wang & Komatsuzaki, 2021) | ✓ | | | | | ? '20 |
| RedPajamas (Computer, 2023) | | ✓ | | 5 ('19-'23) | Mar '23 | Jan '23 |
| Falcon (Almazrouei et al., 2023) | ✓ | | ✓ | | | Feb '23 |
| FalconRW (Almazrouei et al., 2023) | | | ✓ | | | Feb '23 |
| OLMo (Groeneveld et al., 2024) | | ✓ | | 20 ('20-'23) | Mar '23 | June '23 |
| LLaMA (Touvron et al., 2023a) | | ✓ | | 5 ('17-'20) | Aug '22 | ? '20 |

Table 1: Different decoder-only LLMs and their corresponding pre-training data. The CommonCrawl dumps are processed to various degrees. Frequently used datasets with their own columns include RefinedWeb (RW), C4, and the Pile. CC Cutoff indicates the last CommonCrawl dump included, unknown months are marked with a ?.

and have stopped scraping it. Unlike the documents in WIKISPAN, the documents in each bucket have no relation with one another.

## 3.2 Probing Methodology

Our goal is to determine the effective cutoff of the model for a given resource, rather than focusing on individual topics or documents. We do this by measuring perplexity on documents in each month bucket, using the first 512 tokens for Wikipedia and 256 for NYT (due to shorter documents). See Fig. 2 for an example.

**Normalization** Occasionally some documents in our time-spanning datasets have drastically different perplexity compared to previous months (e.g. a Wikipedia page changing to become a redirect page rather than a content page for one month). As this distracts from understanding the relative model perplexity across months, we follow previous work and aggregate perplexities by taking an average of the measured perplexities after discarding the lowest and highest 2.5% of measurements in each month (Shi et al., 2023).

Perplexity measurements are not comparable across unrelated models due to data and training differences. Thus, we normalize the averaged perplexities to a 0-1 scale by performing min-max scaling over the entire time-span in order to compare their fluctuations across time. We call these *relative perplexities*. We take the time at which relative perplexities are minimized to be the effective knowledge cutoff. Since the minima are not always sharp, the effective cutoffs should be interpreted as a distribution over time.

## 3.3 Mining from Pre-training Data

We hypothesize that similar documents in training impact model knowledge and relative perplexity measurements. For example, if parts of a resource were duplicated many times, we might expect perplexity on those documents to be particularly low. Prior work has shown the effects of document frequency on LLM memorization (Carlini et al., 2022).

To better understand the perplexity curves, we search for documents similar to those in our time-spanning datasets. This retrieves old versions of the documents, near duplicates, and copied fragments – all of which may impact information in the model and our perplexity measurements. We expect that the distribution mined from training data is inversely correlated with the perplexity trends – the counts of the versions of the retrieved Wikipedia-alike documents should be higher at the same months where perplexity is lower. The entire process consists of obtaining and indexing nearly 4T tokens from several LLM training sets.

Given a pre-training dataset, we first construct a BM25 index over it using Elasticsearch. For each topic, we use the first 512 words of the document as the query to find similar documents.[5] We use the BM25 scores as a first step filter in finding near duplicates.

---

[5]We use the version of Wikipedia from the model's dump date for queries

Using the top ten BM25 results, we calculate the edit distance between the matched documents and every version of the corresponding Wikipedia topic in WIKISPAN, normalizing by the character length of the matched document to avoid length biases. We classify matched documents as a Wikipedia document only if the minimum of this normalized edit distance score is less than 0.2.[6] With this subset of similar documents, we attribute each document to its closest matching month by edit distance (including ties). This then allows us to plot the ground truth distribution of all documents similar to the original by date. We provide a more detailed description of this algorithm in Appendix A.

## 4 Experimental Setup

### 4.1 Pre-Training Datasets

The LLMs we evaluate were pre-trained on data derived from three major datasets: C4 (Raffel et al., 2020), the Pile (Gao et al., 2020), and RefinedWeb (Penedo et al., 2023), as well as additional CommonCrawl dumps. We describe their contents, as well as how different LLMs used them during training, focusing on subcorporas that may contain Wikipedia or NYTimes content. We note that no open pre-training dataset ever includes a direct dump of NYT articles; they are only present in included CommonCrawl dumps.

**C4** C4 is a single, heavily processed, open-access CommonCrawl dump from April 2019. It uses content-based filters to discard documents containing undesirable content such as obscene words, boilerplate templates, and code. C4 is deduplicated at a three sentence span level, and has an overall size of about 750 GB.

**Pile** The Pile is an open-access curated dataset consisting of 22 sub-datasets and totals around 800GB. The relevant sub-datasets are Pile-CC and the Wikipedia dump. The Pile-CC is deduplicated at a document level, and consists of 22 random chunks out of the 3679 extracted from CommonCrawl dumps between 2013-2020. The Wikipedia dump is from March 2020 and is up-sampled three times.

**RefinedWeb** RefinedWeb (RW) consists of documents from CommonCrawl dumps spanning 2008 to February 2023. Because RW was designed to be used in conjunction with other high quality data sources, URLs from specific top-level domains (including "wikipedia.org") are excluded. The public RW is only a 600B token sample of the total 5T token dataset.

### 4.2 Models

We evaluate a variety of decoder-only Transformer LLMs with accessible (or described) data, as shown in Table 1. We provide speculation about closed-data models in Appendix D, but as we cannot verify correctness, we do not include these results in the main text.

Pythia (Biderman et al., 2023), GPT-Neo (Black et al., 2022), and GPT-J (Wang & Komatsuzaki, 2021) were trained on the Pile, and represented early attempts to replicate closed-access models by the open-source community. The Pythia suite was designed to provide a replicable training process for studying ideas like training dynamics and memorization in LLMs. The RedPajama model suite (Computer, 2023) was intended as an open-access replica of the LLaMA model (Touvron et al., 2023a), providing replicable pre-training data sourced from web-crawls. The Falcon suites are trained on RefinedWeb, a dataset consisting solely of web-scraped data (Almazrouei et al., 2023). Finally, OLMo (Groeneveld et al., 2024), and LLaMA (Touvron et al., 2023a) are the latest generation of LLMs, trained on datasets used by prior models and large amounts of CommonCrawl. These models span several iterations of LLM pre-training approaches (data curation, data size, and model size). Table 1 describes these models in terms of their corresponding training sets and data versions.

---

[6]At most 20% of a document can to be changed to count as an exact match a version of the topic.

## 5   Results

In this section we discuss the results of probing language models to determine their effective cutoffs and compare the dates with the ground truth distributions from their training sets.

### 5.1   NEWSSPAN

As mentioned in Section 3.1, we use NYT data until mid-2020, and thus only evaluate models with pre-2021 CommonCrawl cutoffs due to the lack of evaluation data. Our results are shown in Fig. 3. For each of the models, the perplexity curve increases in early 2020, which is the date of the last CommonCrawl dump included in the Pile. Thus, we see that it is possible to determine the effective knowledge cutoff of different articles posted over time using this method.

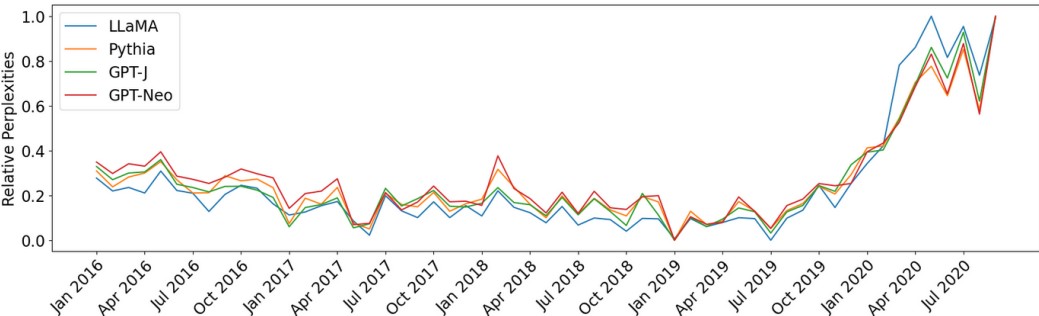

Figure 3: Relative perplexities of models per month using the NEWSSPAN (§3.1) dataset (we use relative as exact perplexities are not needed for determining effective cutoffs). We find that our approach identifies the effective cutoffs as the stated knowledge cutoff for NYT, as models have increased perplexity when their CommnonCrawl data dumps end in 2020.

### 5.2   WIKISPAN

As discussed in Section 4.1, there are three major categories of datasets that models are trained on: C4, the Pile, and Falcon RefinedWeb (RW) and we note that the datasets are only a subset of the training set for some models. Again only uncomparable **relative perplexities** are shown as absolute perplexities between different models are not relevant to our goal of determining knowledge-cutoffs. For each category, we also overlay the computed distribution of similar ground truth documents in light grey (when available) to show the correlation between the ground truth results and effective cutoffs.

**Pile-based Models**   The three models GPT-Neo, GPT-J and Pythia are exclusively trained on the Pile. We show the results of the perplexity measurements in Fig. 4 (upper), where we see a noticeable drop in perplexity around March 2020. This month corresponds exactly to the date of the Wikipedia included in the Pile, indicating the effective cutoff for Wikipedia of the models aligns with the reported Wikipedia cutoff. Appropriately, we also note that the distribution of ground truth versions is highest at that month, corresponding to the 3x up-sampled Wikipedia dump in the Pile.

**FalconRW-based Models**   FalconRW is exclusively trained on RW while Falcon incorporates curated corpora from the Pile. Note that both models were trained on a subset of the RW dataset, the exact subset which is not publicly described. We see in Fig. 4 (middle) that the perplexity curves have a low perplexity from late 2019 to early 2021 with a minimum around January 2020. These results may seem surprising as FalconRW has not seen Wikipedia in training – however, as the overlayed ground truth and Section 6.1 shows, it does contain Wikipedia content found on other top-level domains.

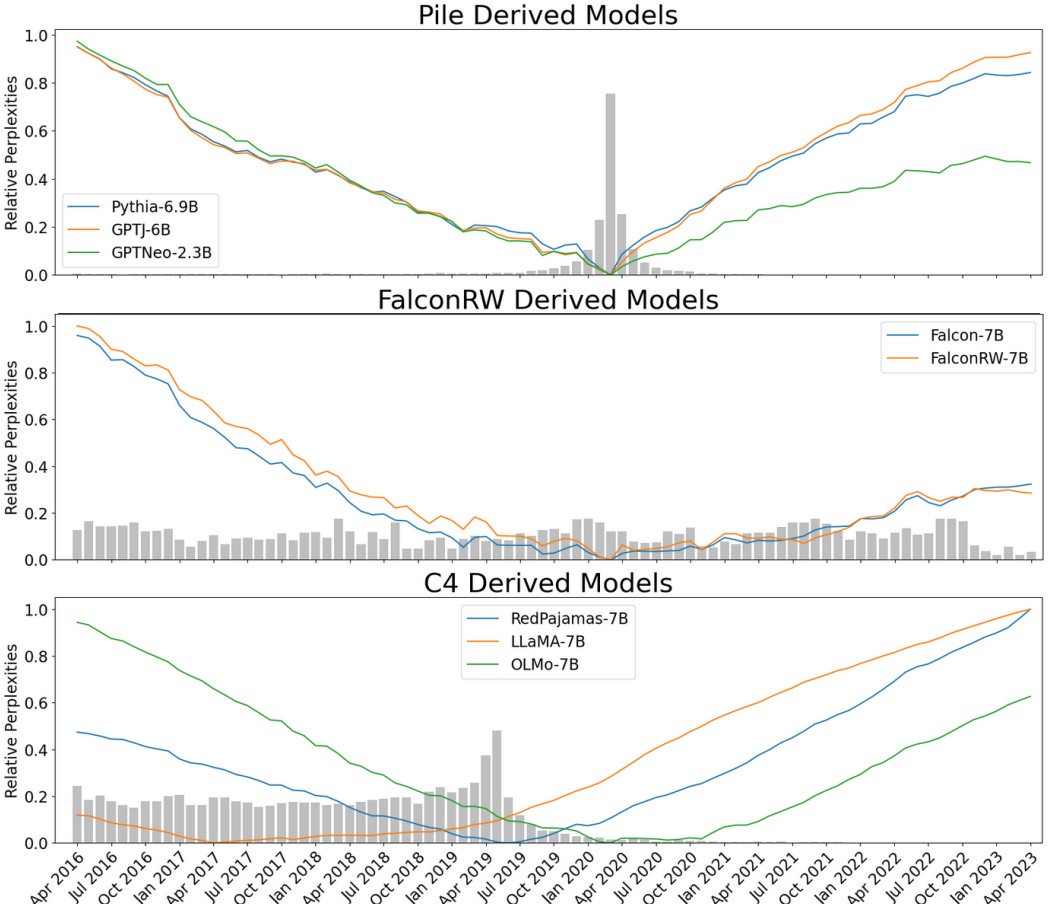

Figure 4: Relative perplexities of models per month using the WIKISPAN (§3.1) dataset. Upper plot shows Pile derived models, middle shows FalconRW derived models, while lower shows C4 derived models. The light grey bars indicate the distribution of Wikipedia-alike documents, matched to their closest version, as calculated in Section 3.3. In some cases the knowledge cutoff aligns with the model's effective cutoff (e.g. the Pile) while more recent models are aligned much earlier (e.g. RedPajamas to 2019, even though it has an explicit 2023 Wikipedia dump).

**C4-based Models** Fig. 4 (lower) shows results for C4-derived models: RedPajamas, OLMo, and LLaMa. While each model uses C4 during pre-training, it only comprises a small portion of their respective training data. The more salient similarity is that each of the models consists of many independent CommonCrawl dumps, and the differences in effective cutoff dates of the three models can be explained by the CommonCrawl dumps included in their training data. LLaMA incorporates 5 dumps from 2017 to 2020, and its cutoff date is thus in that range. In contrast, RedPajamas incorporates 5 dumps from 2019 to 2023, and its effective cutoff is a few months later. OLMo uses all 20 dumps from 2019 to 2023, and thus sees the latest effective cutoff. Nonetheless, the effect of C4 on these models is evidenced by the effective cutoffs being biased towards the C4 CommonCrawl dump date (April 2019). See Section 6.2 for a breakdown of the entire training data of RedPajamas.

**Impact of Scale** We also consider the effect of model size on our methods. We evaluate the perplexity of WIKISPAN under a suite of Pythia (460M, 1B, 2.8B, 6.9B, 12B) and LLaMA (7B, 13B, 65B) models. Fig. 5 shows that while the smaller models have a more varied perplexity curve, they are still minimized at the expected date. This makes intuitive sense as the models are all trained on the same data.

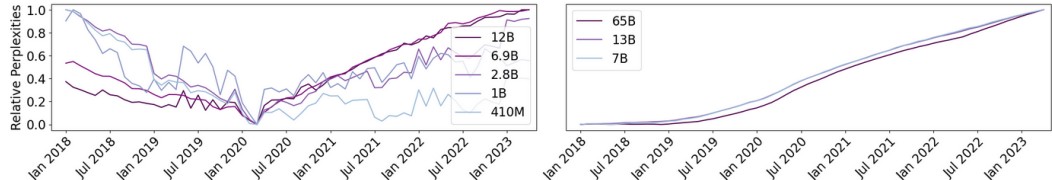

Figure 5: Relative perplexities of models in the Pythia (left) and LLaMA (right) suites. Darker colors indicate larger model size. While the smaller models have a more variable perplexity curve, they are still minimized at the same effective cutoff date.

**Number of Documents**    We lastly consider the effect of the number of documents measured. In some domains, document collection may be difficult; as such, we evaluate our method by varying $x$, the number of documents considered (2500, 1000, 500, 250, 100, 50) for the three C4 derived models. Fig. 6 shows for $x > 50$, the effective cutoffs of the three models are consistent with the full results. $x = 50$ appears to be the threshold where the trend are less consistent, likely due to the increased variance in the data. This shows that our method is robust even when many versions of documents cannot be collected.

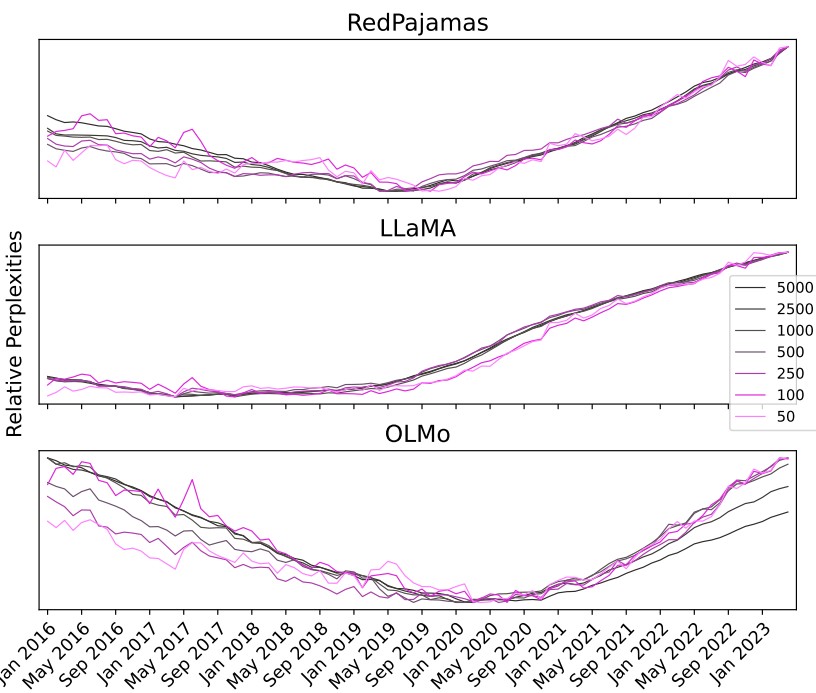

Figure 6: Relative perplexities of RedPajamas (top), LLaMA (middle), and OLMo (bottom) when varying $x$, the number of documents in each bucket. Darker colors indicate more documents, and the black lines corresponding to $x = 5000$ are the results shown in Fig. 4. For small $x$, the perplexity curves are more variable due to the smaller sample size, but for $x > 50$, the ablated results are consistent with the full results.

## 6    Why are models not aligned to their cutoff date?

In this section we describe why a model's effective cutoff and reported cutoff can differ. This mismatch is due to two main factors: (1) deduplication pipelines that ignore semantically equivalent but lexically near duplicates and (2) temporal biases of CommonCrawl dumps.

## 6.1 Complications in Deduplication Pipelines

It is common practice in LLM training pipelines to deduplicate data. In the context of Wikipedia, when a dataset undergoes fuzzy or exact deduplication, one might expect that different versions of Wikipedia pages (near duplicates) and copies of Wikipedia pages (exact duplicates), respectively, are removed from the dataset. However, we empirically find that there exist a large number of near and exact duplicates in training datasets, and provide examples for each. This shows that deduplication pipelines are unable to detect the extra *copies* and *versions* of Wikipedia documents present in CommonCrawl dumps.

**FalconRW**  FalconRW removed all documents that had Wikipedia as a top level domain so they could use FalconRW in conjunction with curated versions in the future (as they did in for the Falcon dataset). They assumed this would deduplicate the data, however, we find that there are still near duplicate Wikipedia documents, as shown in Table 2.

**C4**  C4 was created from one CommonCrawl dump and "discarded all but one of any three-sentence span occurring more than once." However, we show an example in Table 3 of a pair of documents which contain the same three-sentence span. We also observe that the documents are semantically equivalent, and differ only by whitespace characters.

**RedPajamas**  We find that the RedPajamas CommonCrawl dump that was paragraph-level deduplicated contains exact duplicate documents. We show an example in Table 4.

**Discussion**  Out of all the models we evaluated, only Pile derived models exhibit *sharp* alignment towards their reported Wikipedia dump date. This is due to two main factors: the size of its CommonCrawl data (which is minor compared to other models) and the purposeful up-sampling of their Wikipedia dump to match their desired date. The massive amounts of CommonCrawl data that other models are trained on compounds their issues with deduplication, leading to many versions of Wikipedia documents which are not necessarily of the version of their reported dump date.

We confirm this hypothesis by comparing Pythia vs. Pythia-deduplicated. Fig. 7 shows that the deduplicated Pythia, which removes the purposefully up-sampled Wikipedia documents, no longer has the sharp minimum of standard Pythia and instead has an earlier effective cutoff (due to the older CC documents). Thus, we see that the accidental duplicates and the lack of purposeful duplicates (of versions corresponding to the desired effective knowledge cutoff) creates this misalignment in the deduped models.

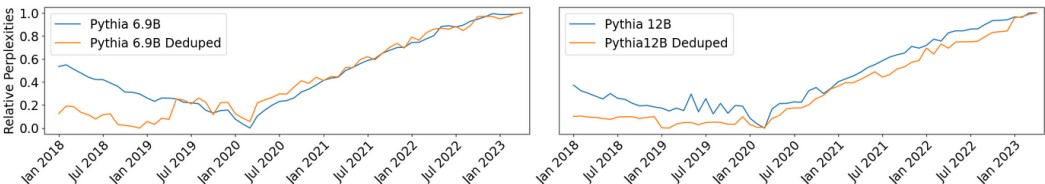

Figure 7: Relative perplexities of models trained on Pile and Pile-dedup. We see that deduplicating the 3x up-sampled Wikipedia in the Pile results in an older temporal alignment due to the included Wikipedia documents from CommonCrawl.

## 6.2 Misalignment of CommonCrawl Dump Dates

All our evaluated models were trained on some portion of CommonCrawl data, with recent models using larger proportions of it. Our ground truth results in Fig. 4 (especially C4-derived models) confirm previous work from Dodge et al. (2021) that suggests a non-trivial amount of data inside of a CommonCrawl dump is actually old data. In the context of Wikipedia, this means that a CommonCrawl dump in 2023 contains many versions of documents dating back to 2016. While models may include a range of CommonCrawl

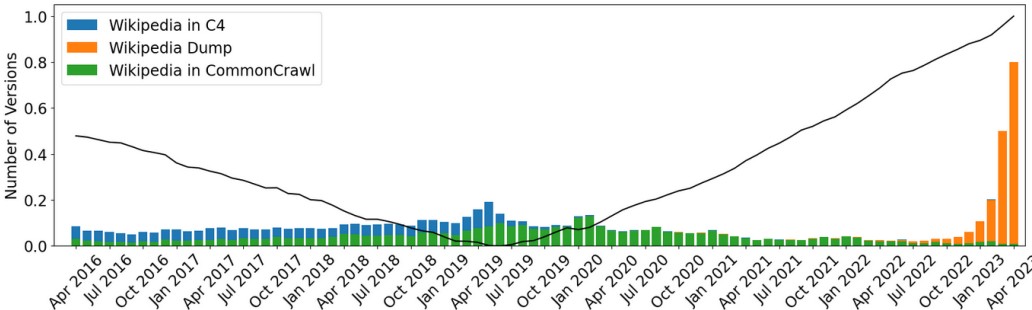

Figure 8: Distribution of Wikipedia versions over the entire RedPajamas training set. Each color represents a different supcorpora of the training set. The black line represents the relative perplexity curve of RedPajamas 7B over WIKISPAN.

dumps, the aggregated data will thus be extremely biased towards earlier dates. To illustrate this concretely for a "newer" style model composed of large amounts of CommonCrawl, we collect the ground truth from all the resources containing Wikipedia in RedPajamas (CommonCrawl, C4, and explicit Wikipedia) and overlay its perplexity curve in Fig. 8.[7]

We see that although the direct Wikipedia dump is included in the pre-training data, over 80% of the Wikipedia documents are from earlier versions (pre-2023). Moreover, versions from the earlier months can and often have duplicate versions as described previously, while the documents in the direct Wikipedia dump are typically not duplicated. We also see that perplexity is minimized around the date of these CommonCrawl Wikipedia versions that compose the majority, in mid-2019.

### 6.3 Summary

In our analysis of available pretraining datasets, we find that CommonCrawl dumps often include *multiple* copies of *different* versions of documents (e.g. Wikipedia). These extra copies and versions are frequently undetected by deduplication pipelines and moreover can consist of outdated information, biasing the effective cutoffs of language models. Thus, we see that there exists two reasons that contribute to the temporal mismatch of a language model's reported and effective cutoff: (1) failures of deduplication pipelines to control for semantic duplicates and (2) the use of newer CommonCrawl dumps to provide updated information when there is a significant amount of older data in the dumps.

## 7 Conclusion

It is now common practice for Large Language Models to provide a "knowledge-cutoff" which intends to communicate to users the date at which LLMs no longer have up to date information. However, this simple metric oversimplifies LLM training in a detrimental manner to usability; it leaves unanswered the questions of "is this knowledge cutoff specific for all resources in the model", "how many copies of my resource are in the model" or "which versions of my corpus are included?" We propose a method to automatically determine the effective cutoff date of LLMs for a given resource and show that although sometimes it does align with the reported cutoff, in many cases it does not. To determine why they fail to align, we analyze the training data of open-data LLMs to discover that there are large quantities of near-duplicates in LLM training data (such as differing only in the citation numbers included in the text) despite efforts from LLM creators to deduplicate. Further, most LLMs rely on CommonCrawl dumps for data, despite the fact that a non-trivial amount of CommonCrawl data is much older than the reported dump date. We hope this analysis will provide insight for users of LLMs who need resource-specific knowledge cutoffs and for LLM-creators who seek to align their LLMs to a given date.

---

[7]Note that although we performed this analysis by binning documents by their most similar version, one can compute an n-gram anaylsis with similar results (Appendix C).

## 8 Acknowledgements

We thank JHU HLTCOE for the support and resources used for this project. We additionally thank the students in JHU CLSP for the valuable feedback, particularly Nathaniel Weir, Kate Sanders, Zhengping Jiang, Jingyu (Jack) Zhang and Aleem Khan. OW is supported by the National Science Foundation Graduate Research Fellowship Program. Any opinion, findings, and conclusions or recommendations expressed in this material are those of the authors(s) and do not necessarily reflect the views of the National Science Foundation.

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

## A   Probing Pseudocode

We denote the pre-training dataset $\mathcal{D}$. Let $M*$ denote the month corresponding to the reported Wikipedia dump in $\mathcal{D}$. Let $D_{T,M}$ represent the version of topic $T$ at month $M$. EDIT refers to the Levenshtein distance between two strings.

---

**Algorithm 1** Counting versions of documents in WIKISPAN

---

1: **procedure** RETRIEVE($\mathcal{D}, M*$)
2:     $counts \leftarrow \{\}$
3:     **for** Topic $T \in$ WIKISPAN **do**
4:         $Q \leftarrow D_{T,M*}[: 512]$                                    ▷ Query with the first 512 tokens
5:         $\mathcal{R} \leftarrow$ BM25$(Q, \mathcal{D})[: 10]$                               ▷ 10 retrieval results
6:         **for** Matched Document $D \in \mathcal{R}$ **do**
7:             $dists \leftarrow []$
8:             **for** Version $V \in D_{T,M_{\text{start}}} \cdots, D_{T,M_{\text{end}}}$ **do**
9:                 $dist \leftarrow$ EDIT$(D, V)/$len$(D)$
10:                 $dists$.append($dist$)
11:             **end for**
12:             **if** $\min(dists) < 0.2$ **then**
13:                 $min\_months \leftarrow$ argmin($dists$)                        ▷ ties identical doc versions
14:                 **for** $m \in min\_months$ **do**
15:                     $counts[m] \leftarrow counts[m] + 1/$len$(min\_months)$
16:                 **end for**
17:             **end if**
18:         **end for**
19:     **end for**
20:     **return** $counts$
21: **end procedure**

---

## B   Deduplication Complications

### B.1   Falcon

---

... By the end of the 17th century, the Chinese economy had recovered from the devastation caused by the wars in which the Ming dynasty were overthrown, and the resulting breakdown of order.[147] In the following century, markets continued to expand as in the late Ming period, but with more trade between regions, a greater dependence on overseas markets and a greatly increased population.[148].[149] The government broadened land ownership by returning land that had been sold to large landowners in the late Ming period by families unable to pay the land tax.[150] To give people more incentives to participate in the market, they reduced the tax burden in comparison with the late Ming, and replaced the corvée system with a head tax used to hire laborers.[151] The administration of the Grand Canal was made more efficient, and transport opened to private merchants.[152] A system of monitoring grain prices eliminated severe shortages, and enabled the price of rice to rise slowly and smoothly through the 18th century.[153] Wary of the power of wealthy merchants, Qing rulers limited their trading licenses and usually refused them permission to open new mines, except in poor areas ...

---

... By the end of the 17th century, the Chinese economy had recovered from the devastation caused by the wars in which the Ming dynasty were overthrown, and the resulting breakdown of order.[148] In the following century, markets continued to expand as in the late Ming period, but with more trade between regions, a greater dependence on overseas markets and a greatly increased population.[149].[150] The government broadened land ownership by returning land that had been sold to large landowners in the late Ming period by families unable to pay the land tax.[151] To give people more incentives to participate in the market, they reduced the tax burden in comparison with the late Ming, and replaced the corvée system with a head tax used to hire laborers.[152] The administration of the Grand Canal was made more efficient, and transport opened to private merchants.[153] A system of monitoring grain prices eliminated severe shortages, and enabled the price of rice to rise slowly and smoothly through the 18th century.[154] Wary of the power of wealthy merchants, Qing rulers limited their trading licenses and usually refused them permission to open new mines, except in poor areas ...

---

Table 2: An example of near-duplicate Wikipedia documents that are semantically equivalent in the FalconRW dataset, differing only by the reference numbers. The documents contain different versions of the Wikipedia article "Qing Dynasty," and are located on lines 168922 and 97669 of the 5th and 3970th parquet files in the public release of FalconRW. The colored text indicates exact matches.

## B.2 C4

Natalie Portman is an actress with dual American and Israeli citizenship. Her first role was as an orphan taken in by a hitman in the 1994 action film Léon: The Professional, but mainstream success came when she was cast as Padmé Amidala in the Star Wars prequel trilogy (released in 1999, 2002 and 2005). In 1999, she enrolled at Harvard University to study psychology while still working as an actress. She completed her bachelor's degree in 2003. In 2001, Portman opened in New York City's Public Theater production of Anton Chekhov's The Seagull. In 2005, Portman won a Golden Globe Award and received an Academy Award nomination for Best Supporting Actress for her performance in the drama Closer. She won a Constellation Award for Best Female Performance and a Saturn Award for Best Actress for her starring role in V for Vendetta (2006). She played leading roles in the historical dramas Goya's Ghosts (2006) and The Other Boleyn Girl (2008). In May 2008, she served as the youngest member of the 61st Annual Cannes Film Festival jury. Portman's directorial debut, Eve, opened the 65th Venice International Film Festival's shorts competition in 2008. Portman directed a segment of the collective film New York, I Love You. Portman is also known for her portrayal as Jane Foster, the love interest of Marvel superhero Thor, in the film adaptation Thor (2011), and its sequel, Thor: The Dark World ... (2013). In 2010, Portman starred in the psychological thriller Black Swan. Her performance received critical praise and earned her a second Golden Globe Award, the Screen Actors Guild Award, the BAFTA Award, the Broadcast Film Critics Association Award and the Academy Award for Best Actress in 2011.

Natalie Portman is an actress with dual American and Israeli citizenship. Her first role was as an orphan taken in by a hitman in the 1994 action film Léon: The Professional, but mainstream success came when she was cast as Padmé Amidala in the Star Wars prequel trilogy (released in 1999, 2002 and 2005). In 1999, she enrolled at Harvard University to study psychology while still working as an actress. She completed her bachelor's degree in 2003.\nIn 2001, Portman opened in New York City's Public Theater production of Anton Chekhov's The Seagull. In 2005, Portman won a Golden Globe Award and received an Academy Award nomination for Best Supporting Actress for her performance in the drama Closer. She won a Constellation Award for Best Female Performance and a Saturn Award for Best Actress for her starring role in V for Vendetta (2006). She played leading roles in the historical dramas Goya's Ghosts (2006) and The Other Boleyn Girl (2008). In May 2008, she served as the youngest member of the 61st Annual Cannes Film Festival jury. Portman's directorial debut, Eve, opened the 65th Venice International Film Festival's shorts competition in 2008. Portman directed a segment of the collective film New York, I Love You. Portman is also known for her portrayal as Jane Foster, the love interest of Marvel superhero Thor, in the film adaptation Thor (2011), and its sequel, Thor: The Dark World ... (2013).\nIn 2010, Portman starred in the psychological thriller Black Swan. Her performance received critical praise and earned her a second Golden Globe Award, the Screen Actors Guild Award, the BAFTA Award, the Broadcast Film Critics Association Award and the Academy Award for Best Actress in 2011.

Table 3: An example of exact three sentence duplicates in C4, along with semantically equivalent text following. The two documents are versions of the Wikipedia article "Natalie Portman." The colored text indicates exact matches.

## B.3 RedPajamas

"Adam Richard Sandler (born September 9, 1966) is an American actor, comedian, screenwriter, film producer, and musician. After becoming a Saturday Night Live cast member, he went on to star in many Hollywood feature films that have grossed over \$2 billion at the box office combined.Sandler's well-known roles include Billy Madison (1995), Happy Gilmore (1996), The Waterboy (1998), The Wedding Singer (1998), Big Daddy (1999), Mr. Deeds (2002), 50 First Dates (2004), The Longest Yard (2005), Click (2006), Grown Ups (2010), Just Go with It (2011), Grown Ups 2 (2013), Blended (2014), and Murder Mystery (2019). He also voices Dracula in the Hotel Transylvania franchise (2012–present). Some of his films, such as the widely panned Jack and Jill, have been heavily criticized, culminating in a shared second place in the number of Raspberry Awards (3) and Raspberry Award nominations (11), in both cases second only to Sylvester Stallone. Sandler ventured into dramatic territory with his roles in Punch-Drunk Love (2002), Spanglish (2004), Reign Over Me (2007), Funny People (2009), The Meyerowitz Stories (New and Selected) (2017), and Uncut Gems (2019), all of which earned him critical praise."

Table 4: An example of the exact document duplicates in RedPajamas. The documents are versions of the Wikipedia article "Adam Sandler," and is duplicated 10 times in the RedPajamas CommonCrawl training data.

## C N-gram Analysis instead of Exact Match

How is perplexity affected when seeing two lexically similar texts? One hypothesis is that perplexity is most affected by exact and near-duplicates. Another hypothesis is that the actual text in a document is factor affecting perplexity. To illustrate the difference between these hypotheses, a document that contains shuffled sentences from a version of a Wikipedia article (shuffling order of sentences) would not affect perplexity in the former case, but would in the latter case. The formeh hypothesis is the basis behind our algorithm for attributing matched documents to their closest versions. We test the latter hypothesis by proposing another way to attribute credit, directly counting the intersection of $n$-grams in matched documents with precomputed sets of $n$-grams sourced from WIKISPAN. We discount by the number of times an ngram appears across all months (similar to an inverse document frequency) in order to count ngrams that are distinct to a specific Wikipedia version. Our algorithm and results are described in Appendix C.1

## C.1 Algorithm Pseudocode

We use the same notation as in [Appendix A](#)

---

**Algorithm 2** Counting n-grams of documents in WIKISPAN

---

1: $ngrams \leftarrow \{\}$
2: **for** Month $m \in$ WIKISPAN **do**            ▷ Compute all n-grams of all docs in month $m$
3:     $ngrams[m] \leftarrow$ Counter($[$NGRAMS($D_{T,m}$) **for** $T \in$ WIKISPAN$]$)
4: **end for**
5: $common\_ngrams \leftarrow \bigcap_{m \in \text{WIKISPAN}} ngrams[m]$
6: **procedure** RETRIEVE($\mathcal{D}, M*$)
7:     $counts \leftarrow \{\}$
8:     **for** Topic $T \in$ WIKISPAN **do**
9:         $Q \leftarrow D_{T,M*}[: 512]$                      ▷ Query with the first 512 tokens
10:         $\mathcal{R} \leftarrow$ BM25($Q, \mathcal{D}$)$[: 10]$                      ▷ 10 retrieval results
11:         **for** Matched Document $D \in \mathcal{R}$ **do**
12:             **for** ngram $n \in$ NGRAMS($D[: 512]$) **do**
13:                 **for** month $m \in$ WIKISPAN **do**
14:                     **if** $n \in ngrams[m]$ **then**
15:                         $counts[m] \leftarrow counts[m] + ngrams[m][n] - common\_ngrams[n]$
16:                     **end if**
17:                 **end for**
18:             **end for**
19:         **end for**
20:     **end for**
21:     **return** $counts$
22: **end procedure**

---

## C.2 Results

We show the perplexity curves of our evaluated models and compare it with our new n-gram statistics.

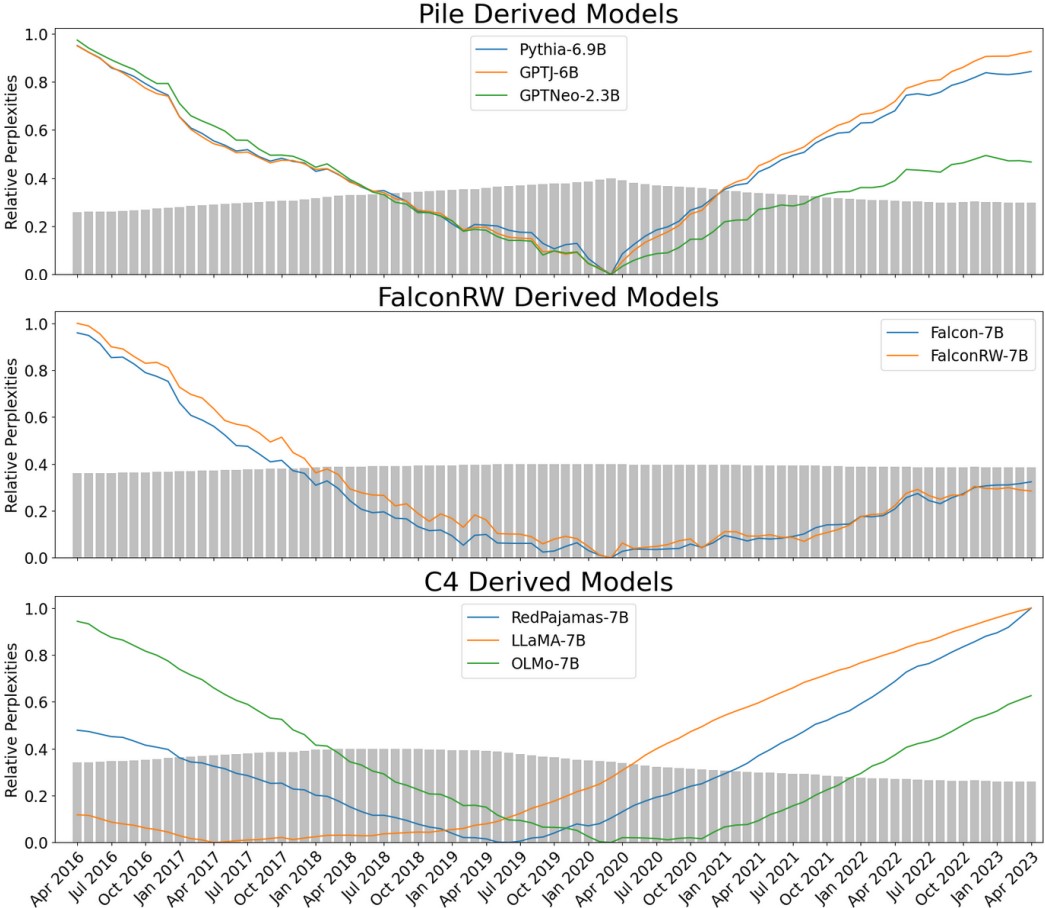

Figure 9: Relative perplexities of models per month using the WIKISPAN (§3.1) dataset (we use relative as exact perplexities are not needed for determining effective cutoffs). Upper plot shows Pile derived models, middle shows FalconRW derived models, while lower shows C4 derived models. The light grey bars indicate the ground truth similar documents, matched to their closest version, as calculated in Appendix C. Note that these datasets are only a subset of the training set for some models. In some cases the knowledge cutoff aligns with the model's effective cutoff (e.g. the Pile) while for more recent models they are aligned much earlier (e.g. RedPajamas to 2019, even though it has an explicit 2023 Wikipedia dump).

# D  Closed Model Results

We evaluate our method on the closed-data models Gemma (Team et al., 2024), LLaMA-2 (Touvron et al., 2023b) and Mistral (Jiang et al., 2023) in Fig. 10. We see that Mistral/LLaMA-2, like many of the open-data models we analyze, has a much earlier effective cutoff for Wikipedia. In contrast, we see that Gemma has a much later effective cutoff, indicating their success at aligning Wikipedia to roughly 2021.

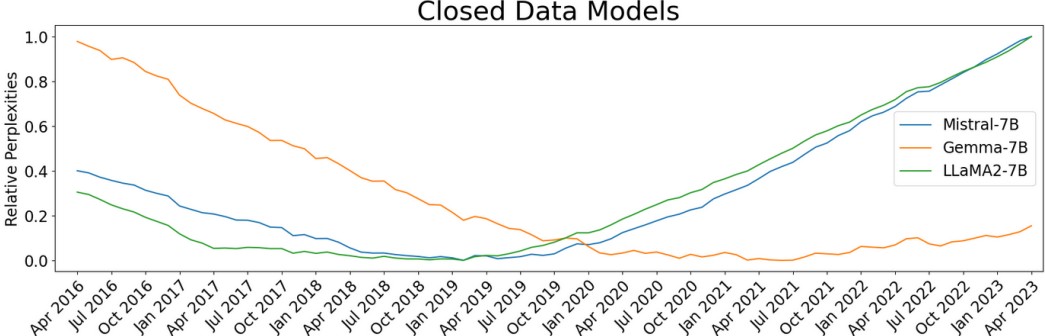

Figure 10: Relative perplexities of models per month using the WIKISPANdataset (we use relative as exact perplexities are not needed for determining effective cutoff).

