# OpenReview forum: "Dated Data: Tracing Knowledge Cutoffs in Large Language Models"
_colmweb.org/COLM/2024/Conference — COLM_

### Official Review · Reviewer_6tHA · 2024-05-09

**Rating:** 9
**Confidence:** 4
**Ethics Flag:** 1

**Summary:**

This paper investigates the concept of knowledge cutoffs in Large Language Models, which is critical for applications requiring up-to-date information. The authors challenge the assumption that a reported cutoff date for an LLM implies uniformity across all training data and propose a method to estimate the effective cutoff at the resource level. Their approach does not require access to the model's pre-training data and reveals significant discrepancies between reported and effective cutoffs. The paper also conducts a large-scale analysis to understand the causes of these inconsistencies, identifying issues with data deduplication and temporal misalignments in CommonCrawl data.

**Questions To Authors:**

- Have you considered or tested the applicability of your approach to other types of resources, such as news articles, books, or legal documents, which may have different update frequencies and versioning challenges?
- How does the model architecture or size influence the identification of effective cutoffs? Would smaller models or those with different training methodologies exhibit similar discrepancies between reported and effective cutoffs?

**Reasons To Accept:**

- The paper introduces a novel method for evaluating the temporal dynamics of knowledge within LLMs. By measuring the perplexity across different versions of datasets, the authors provide a granular understanding of knowledge cutoffs, which is a significant advancement in LLM analysis.

- The large-scale analysis of pre-training datasets is particularly impressive. The identification of issues such as temporal misalignments in CommonCrawl data and deduplication challenges is a substantial finding that contributes to the broader understanding of LLM training data nuances.

- The detailed examination of the reasons behind the misalignment of cutoff dates provides diagnostic value to both LLM creators and users. The insights can guide the development of better training protocols and inform users about the potential limitations of LLM knowledge bases.

**Reasons To Reject:**

- The paper does not discuss the scalability of the proposed method to even larger models or datasets.
- It is unclear how these findings generalize to LLMs trained with different data or methods.

---

> ### Author Rebuttal · Authors · 2024-05-31
>
> Thank you for your review and for finding our work "impressive" and a "substantial finding that contributes to the broader understanding of LLM training data nuances"!
>
> > The paper does not discuss the scalability of the proposed method
>
> We show results for models from 7B to 65B in Figure 5, but agree that we did not test models beyond 65B parameters. We will add more discussion to Section 5.2 discussing this – we believe that the trends from 7B to 65B will continue, as they show consistent results.
>
> As the resource size increases our method will work better as the sample size will be larger and less biased, but we agree that computationally, our method will increase linearly with the size of the dataset.
>
> We will add these comments to Section 4.1 and 4.2.
>
> > How does the model architecture or size influence the identification of effective cutoffs?
>
> Great questions! We evaluate a large variety of language models of varying transformer architectures and training schemes (see Table 1). Moreover, we also evaluate different sized models in Pythia and LLaMA suites (see Figure 5 in section 5.2). Overall, we find that that effective cutoffs are consistent between different sized models that were pre-trained on the same data.
>
> > Have you considered or tested the applicability of your approach to other types of resources
>
> In our paper we showed results for two types of resources (Wikipedia and news articles, as you suggest – that is our NYT collection), as they represent the streaming and updating types of datasets (Section 3). Unfortunately due to paper space we could not add more datasets comprehensively, but our initial tests showed that it holds on other resources as well.
>
> We leave it to future work (and to LLM creators) to document these results for all other resources that users are interested in – there are a lot of resources out there and we agree this is an important area of future work!

---

### Official Review · Reviewer_VAWs · 2024-05-10

**Rating:** 6
**Confidence:** 4
**Ethics Flag:** 1

**Summary:**

This work proposes to reexamine the reported knowledge cutoff dates of LLMs by demonstrating that different "sub-resources" in LLM training might come with different cutoff dates. This work calculates perplexity over a temporally shifting domain corpora to illustrate the perplexity changes corresponding to document time stamps. Experiments found various artifacts of existing models for NYT and Wikipedia domains.

**Reasons To Accept:**

+ this work has the potential to facilitate greater transparency in training data
+ the proposed approach is straightforward

**Reasons To Reject:**

- To verify the actual cutoff date for a certain "sub-resource" or domain, the proposed methodology would need a collection of documents in the domain for a long time period. I wonder how dependent is the approach on those documents: Do we need a lot of documents for each time stamp, or are a few/under 10 often enough? It might be challenging to collect a lot of documents for a domain-specific application and use the proposed method.

- In addition to visualizing the perplexity changes through time, it might be nice to have a methodology to quantitatively decide on a specific cutoff date. In this way, users could decide on whether to trust the LLM when their query concerns information after that specific time stamp. I wonder if the authors might have ideas about how to obtain a specific cutoff date estimation.

- The approach focuses on perplexity: however, it might be the case that certain long-tail documents from a newer date were there in the training data, but were just not properly "memorized" thus leading to high perplexities. I wonder if the authors believe that this might be a confounding factor, and how the proposed methodology mitigates this.

- It would be nice to include a qualitative analysis showing example documents that are before the reported cutoff date but after the "actual" cutoff date, along with perplexity scores/metadata etc.

- After identifying the "actual" cutoff dates of different sub-resources, what could people do with those dates? There could be social aspects such as better issuing a disclaimer about domain-specific knowledge cutoffs, there could also be technical implications such as adapting the model to more recent data on certain domains, instead of training on new data indiscrimnitaly. It would be nice to propose some applications/recommendations that build on this methodology.

---

> ### Author Rebuttal · Authors · 2024-05-31
>
> > Do we need a lot of documents
>
> You bring up an important point about the number of documents required for our method. We empirically found the perplexity of documents in a sub-resource can vary greatly (due to the inherent differences in perplexity in language). As such, we use the median 95% of the perplexity values and take larger samples to obtain more unbiased samples of the true distribution.
>
> Upon your suggestion we are currently running this experiment as we feel it would add to our paper. We are varying the number of docs per timestamp and will provide this in the appendix. It should be done in the next few days and we will post the results here.  Thank you for the suggestion!
>
> > methodology to quantitatively decide on a specific cutoff date
>
> We do provide a baseline approach to quantitatively define the effective cutoff — we use the minima of the perplexity curve, after truncating the distribution to the median 95%. This is what is shown in Figures 3-7. The quantitative way to do this automatically is simply to select the global minima.
>
> > certain long-tail documents from a newer date were there in the training data, but were just not properly "memorized"
>
> This could happen, and we have already attempted to prevent this by using the median 95% of the distribution when calculating values. Thus if the values are on the tails, they will not be included. If there are a large amount of these, then that would be a problem, however, we are unaware of any one collection that contains a large subset of documents that are hard to memorize.
>
> > qualitative analysis … before the reported cutoff date but after the "actual" cutoff date
>
> We included a qualitative analysis of duplicated documents, but agree that some examples of perplexity/docs from before/after the cutoff would be helpful also.  We will update the paper to include examples in a new appendix section!
>
> > what could people do with those dates?
>
> Great question! As discussed in the introduction and conclusion, as an initial step we believe it serves as a warning to both LLM creators and users to be mindful of the temporal biases of LLMs.
>
> For what users could do with that date, it will have to be resource-specific – perhaps if you need the LM up to date, you either pick a different LM or try some techniques to improve its temporal alignment.  However, this paper focuses on finding and notifying the community of this problem, we leave the rest to future work.

---

> ### Author Response · Authors · 2024-06-04
>
> These files in the linked repository show the ablation results where the number of documents in each month bucket are reduced. To be specific, rather than considering the 5000 most edited topics, we repeat the setup described in the paper instead with the x most edited topics. We plot those averages for LLaMA, RedPajamas, and OLMo. Note that when x=5000, this is the same result as in the paper (Figure 4).
>
> We find that for x>50, the effective cutoffs of the three models is consistent with the full results. x=50 is the threshold where the trends and effective cutoffs are less consistent with the original results due to the more apparent variability when taking few samples. Ultimately, we find that these extra results are a useful addition to the paper and confirm our main hypotheses for reasonable number of document bucket sizes.
>
> Thank you for your suggestions and comments!
>
> https://anonymous.4open.science/r/dated-data/README.md

---

### Official Review · Reviewer_eFeV · 2024-05-11

**Rating:** 7
**Confidence:** 4
**Ethics Flag:** 1

**Summary:**

LLMs report a cutoff date which indicates the date till which the LLMs can be considered knowledgeable. This paper introduces the concept of a resource-specific "effective cutoff date," for an LLM. The effective cutoff date, distinct from the "reported cutoff date" of training data for an LLM, is an approximate date of earlier versions of the resource in the training data from which LLMs draw their knowledge. The paper presents an automated technique designed to identify the effective cutoff date for a resource in LLM training data without needing explicit access to training data. Experiments on different LLMs reveal that there are discrepancies between the reported and effective cut-offs that largely stem from two reasons: incomplete deduplication resulting in multiple versions of the same resource in training data and inadvertent leakage of old data in new data dumps.

**Questions To Authors:**

- In section 3.1, under Wikispan, I didn't follow how the documents were slotted into T topics and which topics were filtered out. Why is the topic distinction even necessary?
- In section 3.2, under normalization, it was hard for me to follow how the perplexities were aggregated. What does "average of the median 95% mean"? It would have been helpful to rigorously layout the calculations as math equations to remove any ambiguity.
- In figure 6, the minima in the left panel for both the plots looks to be coinciding. I understand the minima for the dedup curve is not as sharp as for non-dedup, but if they coincide wouldn't the effective cutoff dates be the same with and without deduplication?
- Possible additional citations
  - "Whose Language Counts as High Quality? Measuring Language Ideologies in Text Data Selection" by Gururangan et. al. also discuss the tole of quality filters in training data selection
  - "Speak, Memory: An Archaeology of Books Known to ChatGPT/GPT-4" by Chang et. al. as an example of membership inference testing without the need to compute the LLMs perplexity
- In section 2, under continual learning, the objective of continual learning is a modeling concern which is different from the analytic concern of this paper. What is the more specific connection between the two besides both "examine temporal knowledge"?

**Reasons To Accept:**

The primary strength of the paper is shifting the notion of cutoff dates from a reported cutoff to an empirically calculated effective cutoff. The paper's proposed method for estimating the approximate effective cutoff for a resource is straightforward and easy to replicate. The empirical analysis backs up the thought that the effective date is different from the reported cutoff. A side-effect of the analysis in this paper is that the issues found around data deduplication can serve as valuable insights for further pre-training data curation. The experiments were detailed and provided a clear understanding. Overall, the reporting of a resource-level effective cutoff can increase the transparency of an LLM.

**Reasons To Reject:**

My biggest exception to this paper is the nebulous description of some concepts. Though the paper advances the notion of effective cutoff date, it never formally defines this concept. We're only given a procedure of finding an effective cutoff date and a few examples to illustrate the concept. Similarly, it may not always be clear how one would define a resource within training data and if there is a need to have an effective cutoff date at the resource level or at some other level. In the paper, for example, news from a single media outlet (e.g., nytimes) is considered a resource. But the tax example given in the paper suggests the need for an effective date on a topic rather than a specific news outlet. Moreover, sometimes news articles are sourced and reproduced from common reporting agencies such as Reuters or Associated Press, making it difficult to truly isolate the boundaries of a resource.

---

> ### Author Rebuttal · Authors · 2024-05-31
>
> > My biggest exception … is the nebulous description of some concepts
>
> We will clarify the definition in Sec. 3. We define the effective cutoff date with respect to a model and resource as the date of the version of that resource that most closely aligns with a model. Alignment can be measured in several ways, but we use minimum perplexity.
>
> In the tax example, resource refers to a version of the tax code, not a specific section.  In general, **our approach is for resources (e.g. a set of docs) rather than for attribution to a single doc.**
>
> We agree - as in your news example - that a resource can be more nebulous, and in those cases it would depend on if there are enough unique articles in that resource.  If not, then perhaps it’s not a good resource definition and should be re-defined (as AP news or news broadly).
>
> > I didn't follow how the docs were slotted into T topics
>
> A wikipedia topic is effectively the document title. We use this term to disambiguate between versions of an article and the subject of the article. Wikispan has 94 doc versions (1/month) for each of the 5k topics.
>
> Our pipeline filters more recently created topics as these are blank for certain timespans (e.g. the “Covid-19” page is blank in the year 2017).
>
> > What does "average of the median 95% mean"
>
>
> To remove any outlier ppl values (resulting from unfiltered stub articles/redirects) we first take the mean and 95% confidence interval, also known as a [truncated mean](https://en.wikipedia.org/wiki/Truncated_mean). We will update the wording to make it more clear and include math equations to remove ambiguity!
>
> > if they coincide wouldn't the effective cutoff dates be the same
>
> Interpreting relatively flat basins in the curves indeed requires nuanced considerations. If the two minima coincided, that would be correct. However in Figure 6, while the min of the non-dedup curve coincides with a local minima of the dedup curve, the global min of the dedup curve is around a year earlier.
>
> > connection besides both "examine temporal knowledge"?
>
> That is the connection, as our work explores **if** there is temporal alignment and continual learning explores **how to** aligning them. We will revise this section to make this more clear with our analysis - for example, our analysis could be used “in-the-loop” to determine that a model should undergo additional learning on more recent sources.
>
> We will add the relevant citations!

---

> > ### Comment · Reviewer_eFeV · 2024-06-05
> > **Response**
> >
> > Thanks for the response to the original review. I appreciate the authors taking the effort to engage with all parts of the review.
> >
> > >We will clarify the definition in Sec. 3. We define the effective cutoff date with respect to a model and resource as the date of the version of that resource that most closely aligns with a model. Alignment can be measured in several ways, but we use minimum perplexity.
> >
> > >In the tax example, resource refers to a version of the tax code, not a specific section. In general, our approach is for resources (e.g. a set of docs) rather than for attribution to a single doc.
> >
> > >We agree - as in your news example - that a resource can be more nebulous, and in those cases it would depend on if there are enough unique articles in that resource. If not, then perhaps it’s not a good resource definition and should be re-defined (as AP news or news broadly).
> >
> > The part that the resource is somewhat subjectively decided will remain a point of contention for me although at least giving clear definitions would be useful for future researchers.
> >
> > >A wikipedia topic is effectively the document title. We use this term to disambiguate between versions of an article and the subject of the article. Wikispan has 94 doc versions (1/month) for each of the 5k topics.
> >
> > >Our pipeline filters more recently created topics as these are blank for certain timespans (e.g. the “Covid-19” page is blank in the year 2017).
> >
> > Thanks for this explanation. It makes more sense now.
> >
> > >To remove any outlier ppl values (resulting from unfiltered stub articles/redirects) we first take the mean and 95% confidence interval, also known as a truncated mean. We will update the wording to make it more clear and include math equations to remove ambiguity!
> >
> > Thanks

---

### Official Review · Reviewer_Z85G · 2024-05-14

**Rating:** 9
**Confidence:** 4
**Ethics Flag:** 1

**Summary:**

The paper presents a technique and application for discovering the effective training data cutoff date, via perplexity.

**Questions To Authors:**

no questions

**Reasons To Accept:**

The approach is clever and revealing.  The technique itself is extremely simple and therefore seemingly quite robust.  The experiments demonstrating the success are well-thought out and also very convincing.  The problem is important in LLM accountability, transparency for the user to hold the correct expectations of the model.

**Reasons To Reject:**

None.

---

> ### Author Rebuttal · Authors · 2024-05-31
>
> Thank you for your review and thoughtful comments! We appreciate that you found our work "revealing" and "simple" which were the goals of our analysis to provide transparency for users and creators.

---

### Decision · Program_Chairs · 2024-07-10

**Decision:**

Accept

**Comment:**

This paper introduces the interesting concept of an "effective knowledge cutoff" in LLM training: the date associated with specific resources using during training.  This work exposes the conceptual ambiguity of "reported knowledge cutoffs" as reported by LLM developers by demonstrating how different pre-training sub-collections have different knowledge cutoffs, introduces a simple perplexity-based measure for assessing the effective cutoff of a resource, and carries out several analyses investigating the misalignment between effective cutoffs and reported ones in several LLMs.  Reviewers generally found this work to be creative, conceptually innovative and experimentally robust, with important consequences for data documentation in LLMs.